# Coherent control in the extreme ultraviolet and attosecond regime by synchrotron radiation

Y. Hikosaka [1]*, T. Kaneyasu[2,3], M. Fujimoto[3,4], H. Iwayama[3,4] & M. Katoh [3,4,5]

Quantum manipulation of populations and pathways in matter by light pulses, so-called coherent control, is currently one of the hottest research areas in optical physics and photochemistry. The forefront of coherent control research is moving rapidly into the regime of extreme ultraviolet wavelength and attosecond temporal resolution. This advance has been enabled by the development of high harmonic generation light sources driven by intense femtosecond laser pulses and by the advent of seeded free electron laser sources. Synchrotron radiation, which is usually illustrated as being of poor temporal coherence, hitherto has not been considered as a tool for coherent control. Here we show an approach based on synchrotron radiation to study coherent control in the extreme ultraviolet and attosecond regime. We demonstrate this capability by achieving wave-packet interferometry on Rydberg wave packets generated in helium atoms.

[1] Institute of Liberal Arts and Sciences, University of Toyama, Toyama 930-0194, Japan. [2] SAGA Light Source, Tosu 841-0005, Japan. [3] Institute for Molecular Science, Okazaki 444-8585, Japan. [4] Sokendai (The Graduate University for Advanced Studies), Okazaki 444-8585, Japan. [5] Hiroshima Synchrotron Radiation Center, Hiroshima University, Higashi-Hiroshima 739-0046, Japan. *email: hikosaka@las.u-toyama.ac.jp

Coherence is a fascinating and useful property of light, capable of allowing manipulation of wave functions in matter and of steering populations and pathways. Quantum manipulation with this functional property of light, coherent control, has been demonstrated in various systems from atoms and molecules to solid-state quantum structures[1]. Its applications are rapidly expanding into quantum information science and quantum sensing[2]. Different schemes for coherent control have been invented along with the technological development in optical lasers. One of the most elementary forms of coherent control is the Brumer-Shapiro scheme[3] in which two excitation paths to a resonance are made to interfere using the fundamental and a harmonic of a laser. This basic technique has been followed by more highly developed schemes, among which one standard method is wave-packet interferometry[4,5]. In this scheme, interference between coherent wave packets produced in the system by two laser pulses is controlled by tuning the time delay between them. In addition to nuclear motion guided by femtosecond-scale control, even electron motions have begun to be controlled on the basis of attosecond timing[6].

High harmonic generation (HHG) light generation by intense femtosecond laser pulses enables attosecond coherent control in the extreme ultraviolet (XUV) regime[7,8]. Another fresh light source for wavelengths extending to the X-ray regime is the free electron laser (FEL), which offers extremely high intensity compared to HHG. Although FEL pulses in the self-amplified stimulated emission mode have poor longitudinal coherence, this drawback is effectively removed by electron-beam bunching with an external laser (external seeding). A Brumer-Shapiro-type coherent control in the XUV regime was recently realized with the seeded FEL light source FERMI[9]. While the X-ray range may not be attainable with the external seeding method, self-seeding approaches were already proven to produce near-Fourier-transform-limited X-ray pulses[10,11].

This paper reports another experimental platform to implement coherent control in the XUV and attosecond regime. We demonstrate that synchrotron radiation has a hitherto unremarked ability to achieve coherent control using its intrinsic coherence. While synchrotron radiation from contemporary low-emittance storage rings as normally used possesses a considerable degree of transverse coherence, the random distribution of relativistic electrons in each electron bunch leads unsettled phase relationships between the light wave packets constituting a synchrotron light pulse[12]. The incoherent nature widely attributed to synchrotron radiation results from pile-up of the processes associated with the different light wave packets. Considering that every relativistic electron in an electron bunch travels in an identical magnetic field specifically shaped by a synchrotron device, all the light wave packets generated by the device has basically a common waveform. The concept of the coherent control achieved in this work relies on the use of longitudinal coherence within the waveform of light wave packets produced by individual relativistic electrons, rather than the phase relationships between the light wave packets inside a whole light pulse. Here we report a proof-of-principle experiment demonstrating coherent control in the XUV and attosecond regime by synchrotron radiation.

## Results

**Light wave packet pairs and excitation scheme**. We perform wave-packet interferometry (so-called time-domain Ramsey interferometry[5]) on Rydberg electron wave packets generated in He. The layout of the experiment is sketched in Fig. 1. We have employed two identical undulators installed at a straight section in a storage ring, to produce a tailored light wave packet pair.

Each undulator produces 10-cycle sinusoidal magnetic field to force relativistic electrons to oscillate transversely. Individual relativistic electron traversing inside the twin undulators generate a pair of linearly-polarized electromagnetic wave packets (illustrated in Fig. 1). Each of the light wave packets has a duration of ~1.8 fs and consists of 10-cycle oscillations. The wave packet pair possesses longitudinal coherence between them, which is the essence required to establish coherent control with this wave packet pair. The time delay between the wave packet pair can be increased from original ~2.1 fs, with a phase shifter to lengthen the electron path between the undulators. Light wave packet pairs of this common form, distributing randomly in time, constitute a light pulse from the twin undulators, and the overall pulse duration (300 ps in the present study) corresponds to the electron bunch length.

The radiation spectrum of each undulator was set to have a central photon energy of around 24 eV, where the bandwidth corresponding to the 10-cycle light wave packet was about 10% of the central photon energy. Radiation from the twin undulators irradiated He atoms, after passing through an aperture of 0.4-mm diameter. The broad bandwidth of the radiation covers resonances in He for the excitations of the 1 s electron into different p-type Rydberg orbitals with principal quantum number n of 3-∞ (illustrated in Fig. 2). Two Rydberg wave packets resulting from a coherent superposition of different-n states are thus launched by the interaction of a single He atom with a light wave packet pair. The resultant populations of the Rydberg states can be monitored by following the intensity of fluorescence emitted in their decay pathways. The np → 2 s fluorescence lines in the visible and ultraviolet regions (see Fig. 2), for which a conventional photomultiplier has high sensitivity, were adopted for the present observations. Since the transitions from low-lying Rydberg states are well-separated in wavelength, optical bandpass filters allowed the different lines to be separated easily.

**Time-domain Ramsey interferometry**. The total intensity of fluorescence in the visible and ultraviolet regions, measured as a function of the time delay between the light wave packets, is shown in Fig. 3a. The time spectrum exhibits oscillations of a periodicity around 170 as, and the envelope of the oscillations ramps down and then recovers as the time delay evolves. This spectral feature arises from interferences between the two coherent Rydberg wave packets launched with controlled time delays. Conversely, by tuning the time delay between the light wave packet pair, we can set a particular interference between the Rydberg wave packets and thus control the resultant populations of the Rydberg states, as shown later. Note that the spectral profile in Fig. 3a is determined by the excitations reflecting the undulator spectrum and the state-dependent branching ratios for the fluorescence decays.

The spectral feature seen in Fig. 3a results from overlapping components associated with the populated Rydberg states ($n$ = 3-∞). The contributions from individual Rydberg states can be extracted by selecting the wavelength for fluorescence detection. The spectra in Fig. 3b are obtained in that way for the np Rydberg states ($n$ = 4–6), delineating the delay-dependent populations of these states (The spectra for 3p and 7p&8p are shown in supplementary fig. 1). These spectra exhibit clear fringe features with oscillation frequencies slightly different from each other. The best fits to these features with a sinusoidal function convoluted by Gaussian are shown by solid lines.

The fringe features on these time spectra are created as follows[13]. The first light wave packet produces a superposition of the ground and each Rydberg state in He, oscillating at the transition frequency. Then the second wave packet coherently overlaps another

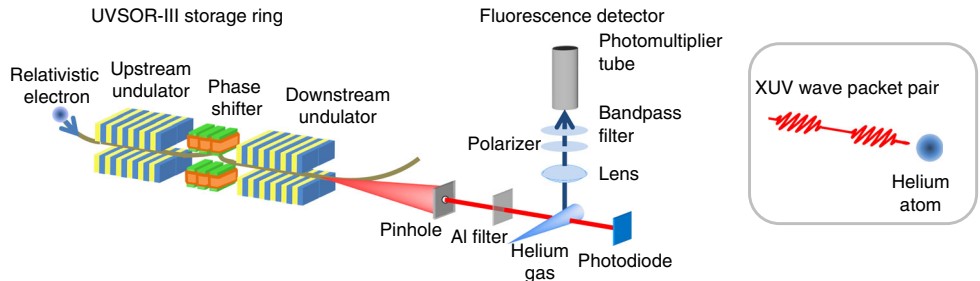

**Fig. 1** Schematic layout of the experimental setup. Individual relativistic electrons traversing inside the twin undulators generate a pair of linearly-polarized electromagnetic wave packets. Each of the light wave packets has a duration of ~1.8 fs and consists of 10-cycle oscillations. Some $10^9$ of light wave packet pairs with this common waveform, distributing randomly in time, constitute a light pulse (300-ps duration) from the twin undulators

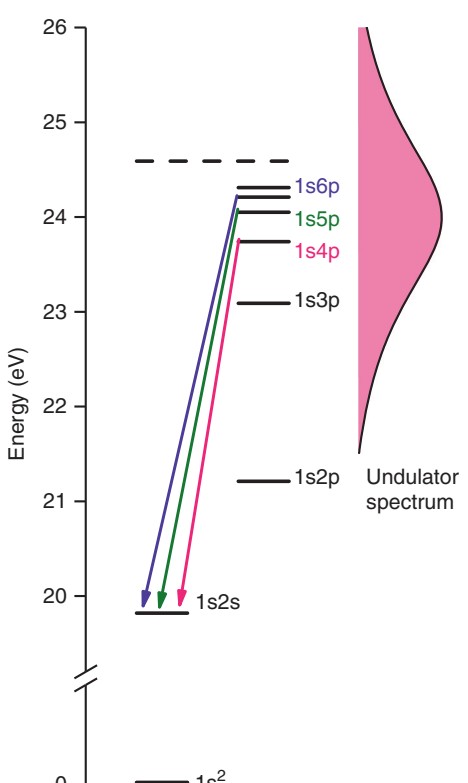

**Fig. 2** Energy level diagram of He states. Fluorescence transitions selected in the measurements of the spectra in Fig. 3b are presented. The undulator spectrum has a bandwidth of about 10% of the central photon energy of around 24 eV, which covers resonances for the excitations of a 1 s electron into many p-type Rydberg orbitals

observation proves that this concept is applicable even with synchrotron radiation. This high-resolution approach without any monochromator will have a potential for expansion in synchrotron sciences.

**Population control**. The time spectra in Fig. 3b, modulating with the individual frequencies, manifest that the relative intensities of fluorescence from these Rydberg states are dependent on the time delay of the light wave packet pair. The relative intensities sampled at four particular time delays are represented in Fig. 4. It is clearly demonstrated here that the fluorescence intensities, reflecting the populations of these Rydberg states, strongly vary with the time delay. For instance, all the Rydberg states are favorably and unfavorably formed at time delays of 80 as and 165 as, respectively. In another case, the formation of the 4p Rydberg state is enhanced at 1515 as, whereas its formation is strongly suppressed at 1590 as. This observation manifests that we can control the populations of the Rydberg states by tuning the time delay between the light wave packets. While this sort of coherent control based on wave-packet interference has been performed with optical laser pulses[16], the platform with synchrotron radiation dramatically expands the range of possible targets toward shorter wavelength than ever achieved before[15].

The fringe intensities seen in Fig. 3b do not completely drop to zero at the valleys. Fitting with convoluted sinusoidal function to these fringe features shows Gaussian widths of 20 ± 1 as (standard deviation). Apart from the temporal resolution of the present interferometry, unevenness in intensities of the two undulator spectra and stray synchrotron light from bending magnets can contribute to the obtained Gaussian widths. The temporal resolution of the present interferometry, which should be thus better than 20 ± 1 as, is broadly comparable to the best values achieved by optical lasers[17] and a seeded FEL[9]. The high temporal resolution of the present interferometry relies on the accurate motion of individual relativistic electrons in the well-defined magnetic field of the synchrotron devices. To put it differently, such fringe contrast can be used as a stringent probe for operating conditions of synchrotron accelerators and insertion devices.

## Discussion

This work has demonstrated that synchrotron radiation from the twin undulator setup provides an original framework for coherent control in the XUV and attosecond regime. The concept of the coherent control demonstrated in this work relies on the use of longitudinal coherence within a light wave packet pair produced by individual relativistic electrons, rather than temporal coherence of a whole light pulse. The temporal resolution achieved with this coherent-control concept is completely unrelated to the bunch length and is determined by the accuracy and uniformity of the waveforms of light wave packets. Synchrotron radiation

oscillation of the same frequency, at the time delay we set. In-phase and out-of-phase overlaps of these oscillations make the peaks and valleys, respectively, in the time spectrum for the Rydberg state. The oscillation frequencies on the time spectra, therefore, should rigorously agree with the transition frequencies to the Rydberg states, regardless of the observed decay channels. The scale of time delay was actually calibrated with the oscillation frequency observed in the spectrum for the 6p Rydberg state. With this calibration, the oscillation frequencies estimated by the fittings to the time spectra for the 4p and 5p Rydberg states deviate only ~$2 \times 10^{12}$ Hz (~9 meV) from the true values. The accuracy can be easily improved much further by measuring over a longer time range. Ramsey interferometry as high-resolution spectroscopy in the XUV range was already established in the HHG basis[14,15], and the present

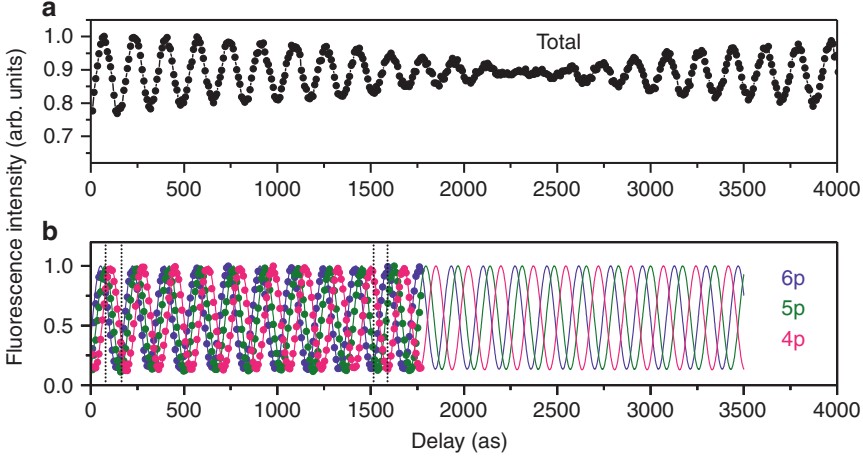

**Fig. 3** Intensities of fluorescent photons from He. Fluorescence intensities are measured as a function of the time delay between the light wave packets: **a** total fluorescence in the visible and ultraviolet regions and **b** contributions from the p-type Rydberg states of $n = 4$–6. The fluorescence intensities were normalized by the intensities of the undulator radiation and then multiplied so that the maxima become unity. The statistical error of each point in the spectra is in 1–2% of the intensity. The total fluorescence spectrum in **a** includes a constant background and the baseline is around 0.62 (see supplementary note 1). Delay time of the horizontal axis is measured from the original delay (~2.1 fs) between the light wave packets, thereby, for the absolute time delay, ~2.1 fs has to be added to the present scale. The scale of the time delay was calibrated with the oscillation frequency observed in the spectrum for the 6p Rydberg state. In **b**, while the experimental values (dots) for the individual Rydberg states were measured until a time delay of 1770 as, the best fits to the experimental values with a sinusoidal function convoluted by Gaussian are depicted with solid lines continuing up to a time delay of 3500 as. The time delays chosen for sampling the Rydberg populations plotted in Fig. 4 are indicated in **b** with vertical lines

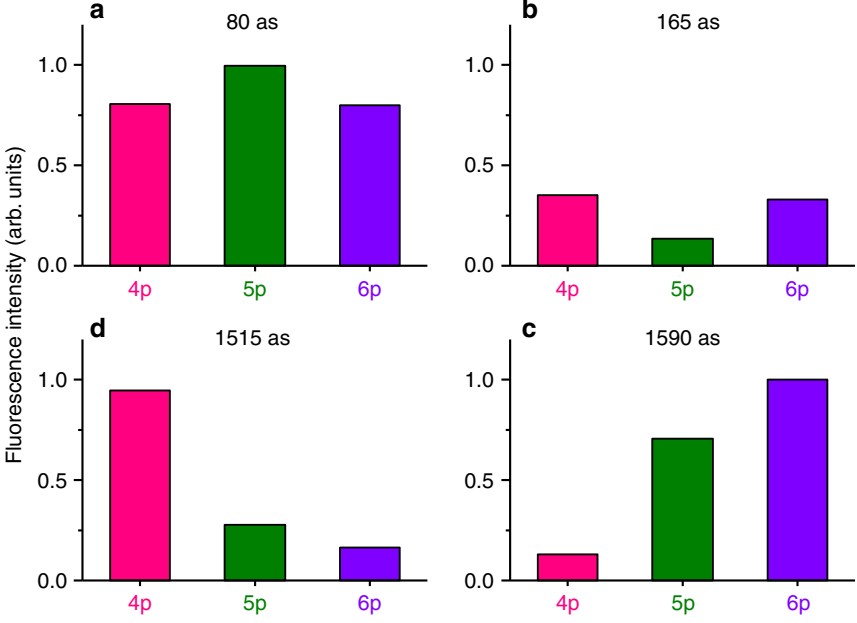

**Fig. 4** Populations of Rydberg states. Populations of 4p, 5p, and 6p Rydberg states in He at four different time delays between the light wave packets, extracted from the Ramsey-fringe spectra in Fig. 3b. The statistical error of each value is in 1–2%

possesses a decisive superiority to HHG and seeded FEL, in its continuous and easy tunability extending as far as the hard X-ray range. There is no essential restriction on the application of the present coherent control concept to such shorter wavelength. Beyond the present prototypical experiment, the coherent-control framework is thus capable of wide application to varieties of quantum processes in atoms, molecules, and condensed matter. Particularly in the X-ray regime, processes involving core electrons can become new targets for coherent control, where the attosecond temporal resolution of the present method will play an active and effective role. Different polarization can be easily generated by undulators, which is a useful option in coherent

control by synchrotron radiation. In fact, we have most recently succeeded in superposing magnetic sublevels in a controlled way using circularly-polarized light wave packets. Besides coherent control, the concept of coherent wave packet pair from a twin undulator setup has enormous potentialities. As already mentioned, Ramsey interferometry with coherent wave packet pair can be used for high-resolution spectroscopy in XUV and also X-ray range, and applied as a unique diagnostic method for electron beams in synchrotron accelerators. We believe that, while the present demonstration is quite limited, the further development of this unexploited capability will advance the frontier of synchrotron sciences.

## Methods

**Experimental setup.** The experiments were carried out at the undulator beamline BL1U of the UVSOR-III storage ring in Okazaki, Japan. The natural emittance of the 750-MeV storage ring is as low as 17.5 nm-rad, leading a favorable transverse coherence property in the XUV range, and the natural bunch length is 300 ps (FWHM). The layout of the experimental setup is shown in Fig. 1. Two 10-cycle APPLE-II undulators were installed in a straight section of the storage ring, and both were operated in the linear polarization mode. The electron path between the undulators can be lengthened by a phase shifter consisting of three pairs of electromagnets and thereby forming a small chicane for the electron beam.

An aperture of 0.4-mm diameter, to select the central part in the cross section of the undulator radiation, was placed 8.2 m from the downstream undulator. An aluminum filer of 75-nm thickness was inserted downstream of the aperture, to eliminate stray light in the visible and ultraviolet ranges. The skimmed undulator radiation illuminated sample gas admitted in the form of effusive beam from a needle of 0.2-mm inside diameter. The interaction point was located around 2 m downstream of the aperture. The effective gas pressure at the interaction point was maintained below $1 \times 10^{-3}$ Pa.

**Fluorescence detection.** Fluorescence photons in the visible and ultraviolet regions were detected at right angle to both the polarization and propagation of the undulator radiation, by a photomultiplier tube (Hamamatsu R6249P). A quartz lens (Sigmakoki SLSQ-30B-30P) provided collection of fluorescence over 0.6-sr solid angle. A polarizer (Thorlabs WP25M-UB) aligned to the light polarization was inserted, to reduce contamination from scattered light. To select fluorescence lines, one of three bandpass filters (Edmund Optics 394nm-CWL, Asahi Spectra HQBP360-UV & HQBP350-UV) was installed in front of the photomultiplier. The fluorescence count rate at the peaks in the Ramsey fringes was 1300–4500 cps, with the storage ring operating at 6–12 mA electron-beam current. The time spectra in Fig. 2 were measured with 4 s accumulation at each point, and normalized to the intensity of the undulator radiation. The undulator radiation intensity was monitored by a Si photodiode (IRD AXUV100G) behind the interaction point.

## Data availability

The data that support the findings of this study are available from the corresponding author upon reasonable request.

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

## Acknowledgements

The authors thank J.H.D. Eland (Oxford) for useful discussions and his careful reading of the manuscript. This work was supported in part by JSPS KAKENHI (grant numbers 17H01075, 18K03489, and 18K11945). The construction of BL1U at UVSOR was supported by the Quantum Beam Technology Program of MEXT/JST.

## Author contributions

Y.H. and T.K. designed the experiment. Y.H., T.K., M.F, H.I and M.K. carried out the measurements, and T.K. analyzed the data. Y.H., T.K. and M.K. discussed the results and interpretation. The manuscript was drafted by Y.H. and completed in consultation with all authors.

## Competing interests

The authors declare no competing interests.
