## [Peer Review File · Nature Communications]

Reviewers' comments:

Reviewer #1 (Remarks to the Author):

The authors report on a new technique for coherently controlling the populations of excited states of helium. The technique relies on using two identical undulators to produce pulse replicas in the extreme ultraviolet whose relative delay can be accurately controlled using a phase shifter. By varying the delay between the pulses oscillations in the total fluorescence signal from helium are observed. The frequency content of these oscillations is analysed using a selection of spectral filters.

The experiment, although conceptually easy to understand, represents a new approach for controlling the populations of excited states in atoms. As pointed out by the authors synchrotrons have the advantage of being widely tuneable offering the prospect of extending this technique to spectral regions which are difficult to achieve with alternative light sources.

However, there were a few areas which were unclear in the paper. In order to be suitable for publication the authors should address the following points:

1. On page 2 the authors state that coherent control by synchrotron radiation can only be achieved through "clever use of the longitudinal coherence". However, the precise details on how this is achieved are lacking in the paper. In Phys. Rev. Accel. Beams 20, 110702 (2017) the authors discuss in detail the relationship between bunch duration and temporal coherence. The wavelength used in this experiment is considerably shorter than the bunch length at UVSOR, which should limit the temporal coherence which can be achieved. Can the authors explain in more details the mechanism by which high temporal coherence is increased as well as the limits of this technique as it is extended to different spectral regions?
2. The pulse energy is sufficient to excite from the ground state to p-orbitals with $n=3-\infty$ states. However, the spectral analysis of the fluorescence has been limited to 4p,5p,6p. Why has the 3p been excluded from the analysis?
3. The time series of the total fluorescence measurement in Figure 2 (a) extends to 4000 as. However, the measured data in Figure 2 (b) only extends to 1770 as. Can the authors indicate in the paper why this is the case?
4. Can the data from the 4p,5p,6p measurements be added and compared with that in Figure 2 (a). It appears in Figure 2 (b) that they contribute approximately in phase at early delays and are falling out of phase towards the end of the measurement range. It would be informative to see how their sum matches the total measured fluorescence. If good agreement is observed it would also highlight that the states which were analysed were the dominant contributions.

Reviewer #2 (Remarks to the Author):

Comments on manuscript "Coherent control in the extreme ultraviolet and attosecond regime by synchrotron radiation" by Y. Hikosaka et al.

Hikosaka et al. report a study where they employ a delayed light pulse pair from a synchrotron to excite a coherent superposition of p-type Rydberg states in Helium just below the ionization potential. They find that the VIS/UV fluorescence intensity oscillates with a frequency given by the energy level separation of the Rydberg states, and reconstruct the populations of the Rydberg states from the oscillations.

While it is of course generally interesting that such fast dynamics can be measured with synchrotron

radiation, I have a number of concerns, mostly concerning the overall lack of information in the manuscript, but also concerning the overall applicability of their measurement technique.

- The reader is supposed to know many concepts and they are never really explained, like Ramsey interferometry. The authors should explain these things better.
- Claims are made in the abstract that seem odd, such as that synchrotrons are incoherent light sources. Synchrotrons are rather known for remarkable spatial coherence, which is why they are being used for coherent diffractive imaging. They authors mean a different form of coherence, but then such a general comment cannot be made in the abstract.
- It is never explained why the fluorescence intensity in the UV/VIS oscillates with the frequency of an XUV wave packet.
- The spectra of the fluorescent lines in the UV/VIS which are used for detection are never shown.
- The time axis in Fig. 2 has his zero at 2.1 fs. Why? And why were the oscillations in Fig. 2b not scanned until zero (i.e. 2.1 fs).
- There are no error bars in the entire manuscript. The authors make a rather hand waving estimate of a time resolution of 20 as.
- No level diagram is shown, of what is excited, and which transitions are fluorescing.
- I am rather concerned with the generality of this technique; the following questions should be clarified:
 - o What was the pulse duration of the synchrotron light pulses. The authors talk about light wave-packets with pulse duration of 1.8 fs. This cannot be the pulse duration of the pulses from the synchrotron. A pulse duration measurement should be provided.
 - o How was the time delay calibrated?
 - o What limits the time resolution in this experiment? I suspect that the synchrotron pulses are long, and the wave packets can only be detected because the lifetimes of the fluorescent states are even longer than the pulse durations. Is yes, this severely limits the general applicability.

The authors should address the above points, and I encourage the authors to write a detailed supporting document, or extend the paper significantly. Beforehand, I do not dare to make a final assessment on the paper.

Reply to the comments from Reviewer 1:

Thank you very much for your time and effort for reviewing our manuscript. Your comments are all useful and we have carefully revised through the manuscript accordingly. In the revised manuscript, the sentences added/modified are shown in red. We decided also to provide a supplementary information file, to present additional information. We believe the revision succeeded in clarifying the value of our work, and we hope you agree on it.

Comment: 1. On page 2 the authors state that coherent control by synchrotron radiation can only be achieved through “clever use of the longitudinal coherence”. However, the precise details on how this is achieved are lacking in the paper. In Phys. Rev. Accel. Beams 20, 110702 (2017) the authors discuss in detail the relationship between bunch duration and temporal coherence. The wavelength used in this experiment is considerably shorter than the bunch length at UVSOR, which should limit the temporal coherence which can be achieved. Can the authors explain in more details the mechanism by which high temporal coherence is increased as well as the limits of this technique as it is extended to different spectral regions?

Answer: As verified in the PRAB paper, temporal coherence of the light pulse from a bending magnet can be improved by restricting the spectral range of the light. The light pulse from an undulator has already a narrow spectrum (similar to the band-passed bending magnet light in the PRAB paper), determined by the magnetic periods of the device, and the corresponding coherence time is typically in the order of femtosecond. The coherence time is much shorter than the light pulse duration according to the electron bunch length (300 ps for our synchrotron). This situation is understood as that many light wave packets of fs duration distribute randomly in time and constitute a light pulse of picosecond duration. This concept, light wave packets from undulator, is widely accepted; for example, it is instructively written in the introduction of Phys. Rev. A, 80 063804 (2009).

The principle of the coherent control achieved in this work relies on the use of longitudinal coherence within a light wave packet pair, rather than temporal coherence of a whole light pulse. Considering that every relativistic electron in an electron bunch travels in an identical magnetic field specifically shaped by a synchrotron device, all the light wave packets generated by the device has basically a common waveform (like the one illustrated in the right panel of Fig.1). We propose in this paper that coherent control by synchrotron radiation can be established by the use of the longitudinal coherence existing naturally in the waveform of the light wave

packets. While temporal coherence of a whole light pulse is limited by the electron bunch duration (300-ps duration at UVSOR), the common waveform that the light wave packets have is essentially independent of the bunch duration. Thus, the temporal resolution achieved with the present coherent-control concept is completely unrelated to the bunch duration and is determined by the accuracy and uniformity of the waveforms of light wave packet pairs. On the application in short-wavelength range, the light wave packet pairs become to have higher-frequency oscillations, but still the longitudinal coherence in the well-defined waveform is unaffected. Therefore, there is no essential restriction on application of the present coherent control concept to shorter wavelength. Here, the time range of target process is an interesting factor. In the present study, we observed nanosecond fluorescence in He, which is much longer than the time scale of delay between the light wave packets. If we targeted a decay process much shorter than the minimum delay of 2.1 fs, the two wave packets launched inside the matter by the light wave packet pair could not interfere to each other and we would see no oscillation in the time spectrum. Most interesting case is that the decay lifetime is comparable to the time range we can scan (i.e. femtosecond range), where the envelope of the interference pattern will reflect the lifetime of the target state. Then one may consider that core-hole processes in x-ray range are too fast to be targeted. This may be true, but subsequent processes after the core-hole decays can be in the observation range. At present we cannot grasp how it appears on the time spectrum, and are interested in such study. We have already started the investigations to He doubly-excited states and Xe 4d resonances with a similar experimental scheme and a more advanced set-up (photoelectron spectroscopy), and publishable data will be gained soon.

To clarify the coherent control principle, we inspected manuscript through and added/modified many sentences, particularly in pages 2&4.

Comment: 2. The pulse energy is sufficient to excite from the group state to p-orbitals with $n=3-\infty$ states. However, the spectral analysis of the fluorescence has been limited to 4p,5p,6p. Why has the 3p been excluded from the analysis?

Answer: The time spectrum for 3p is shown in the supplementary information attached to the revised manuscript. The statistics of this spectrum (5% error) is worse than those of other spectra (1-2% error), owing to low count rate and large background contribution (which was subtracted already in the shown spectrum). In the figure, we plotted also the spectrum measured for 7p&8p (the two fluorescence lines could not be resolved with the band-pass filter used). Figure 3(b) in the manuscript is already crowded with the three time spectra and the fits, and thus we would like to leave the spectra for 3p and 7p&8p in the supplementary information and just to note it in the manuscript.

Comment: 3. The time series of the total fluorescence measurement in Figure 2 (a) extends to 4000 as. However, the measured data in Figure 2 (b) only extends to 1770 as. Can the authors indicate in the paper why this is the case?

Answer: It is simply due to the limitation of experimental time, and we made long measurement only to the total fluorescence and measurements with band-pass filters were stopped around 1770 as. To indicate it better, we modified the figure caption as “while the experimental values (dots) for the individual Rydberg states were measured until a time delay of 1770 as....” .

Comment: 4. Can the data from the 4p,5p,6p measurements be added and compared with that in Figure 2 (a). It appears in Figure 2 (b) that they contribute approximately in phase at early delays and are falling out of phase towards the end of the measurement range. It would be informative to see how their sum matches the total measured fluorescence. If good agreement is observed it would also highlight that the states which were analysed were the dominant contributions.

Answer: In the supplementary information we decide to provide, the sum of the spectra for 3p,4p,5p, 6p, and 7p&8p is compared with the total fluorescence spectrum. The total fluorescence spectrum is well reproduced when the spectra for 3p and 7p&8p were multiplied by a factor of 0.5 and 2, respectively, before the summation. Thus, contributions from 4p-8p are comparable in the total fluorescence, while that from 3p is about half. These contributions are basically determined by the undulation spectrum for their excitations and the state-dependent branching ratios for the fluorescence decays. The ramp-down seen around 2500 as in the total fluorescence spectrum is insufficiently reproduced in the sum spectrum, suggesting non-negligible contributions from higher-n states.

Reply to the comments from Reviewer 2:

Thank you very much for your time and effort for reviewing our manuscript. Your comments are all useful and we have carefully revised through the manuscript accordingly. In the revised manuscript, the sentences added/modified are shown in red. We decided also to provide a supplementary information file, to present additional information. We believe the revision succeeded in clarifying the value of our work, and we hope you agree on it.

Comment: - The reader is supposed to know many concepts and they are never really explained, like Ramsey interferometry. The authors should explain these things better.

Answer: We have carefully inspected the manuscript though, and we added explanations to specialized words or replaced them. As the referee suggested, we used “Ramsey interferometry” or “Ramsey fringe” as usually used in special journals, but found that these technical words do not add any useful information and rather give confusion to general readers. In the revised manuscript, we simply write “time-domain interferometry” and “fringe structure”.

Comment: - Claims are made in the abstract that seem odd, such as that synchrotrons are incoherent light sources. Synchrotrons are rather known for remarkable spatial coherence, which is why they are being used for coherent diffractive imaging. They authors mean a different form of coherence, but then such a general comment cannot be made in the abstract.

Answer: We wrote in the abstract that synchrotrons are incoherent light sources, which is about temporal coherence, not about spatial coherence. To clarify it, we write in the revised manuscript, “Synchrotron radiation, which is usually illustrated as being of poor temporal coherence, hitherto has not been considered as a tool for coherent control.” In the meantime, as the referee points out, we should clearly note the fact that synchrotrons have remarkable spatial coherence. Accordingly, we modified a sentence in page 2 left as “While synchrotron radiation from contemporary low-emittance storage rings as normally used possesses a considerable degree of transverse coherence, the random distribution of relativistic electrons in each electron bunch leads unsettled phase relationships between the light wave packets constituting a synchrotron light pulse”.

Comment: - It is never explained why the fluorescence intensity in the UV/VIS oscillates with the frequency of an XUV wave packet.

Answer: We intended to explain the formation mechanism of the Ramsey fringe in page 3 right,

but it would not be clear enough. We added the words “regardless of the observed decay channels” after “The oscillation frequencies of the Ramsey fringes, therefore, should rigorously agree with the transition frequencies to the Rydberg states”. We hope these words clarify the situation.

Comment: - The spectra of the fluorescent lines in the UV/VIS which are used for detection are never shown.

Answer: We used optical band-pass filters to select the fluorescent lines, and no fluorescence spectrum to be shown is available. However, readers must have the same question. To clarify our use of the optical band-pass filters, we changed a sentence around page 2 end: “Since the transitions from low-lying Rydberg states are well-separated in wavelength, optical band-pass filters allowed the different lines to be separated easily.”.

Comment: - The time axis in Fig. 2 has its zero at 2.1 fs. Why? And why were the oscillations in Fig. 2b not scanned until zero (i.e. 2.1 fs).

Answer: Each relativistic electron has to travel a path between the two undulators, which takes about 2.1 fs. We can only increase the delay with a phase shifter to lengthen the electron path, and thus the minimum is 2.1 fs. The scale in Fig.2 (Fig.3 in the revised manuscript) is measured from the minimum (i.e., 2.1 fs) and 2.1 fs has to be added to get the absolute time delay. We realized the corresponding sentence in the figure caption is somehow misleading and changed the sentence to “for the absolute time delay, ~2.1 fs has to be added to the present scale.”.

Comment: - There are no error bars in the entire manuscript. The authors make a rather hand waving estimate of a time resolution of 20 as.

Answer: We put the error (± 1 as) to the time resolution. The statistical error of each point in the time spectra of Fig.3 and that of each value in Fig. 4 are in 1-3% of the intensity, which is newly noted in the figure captions.

Comment: - No level diagram is shown, of what is excited, and which transitions are fluorescing.

Answer: A level diagram is added as new Fig. 2, where the broad bandwidth of the undulations and the decay transitions are indicated. We agree that this figure is useful for readers to grasp the processes we observed.

Comment: I am rather concerned with the generality of this technique; the following questions should be clarified: What was the pulse duration of the synchrotron light pulses. The

authors talk about light wave-packets with pulse duration of 1.8 fs. This cannot be the pulse duration of the pulses from the synchrotron. A pulse duration measurement should be provided.

Answer: We forgot to note the light pulse duration which is 300 ps (FWHM). This value is noted in page 2 right and also in Methods. The duration of each light wave-packet is 1.8 fs, which is estimated by a theoretical calculation of relativistic electron radiation.

Comment: How was the time delay calibrated?

Answer: Since the oscillation frequency should be the transition frequency to each resonance, the scale of time delay was calibrated with the oscillation frequency observed in the spectrum for the 6p Rydberg state, and the scale was adapted to other spectra. While it was stated around page 3 end, we realized the way of calibration should be noted also in the caption of Fig.3. Thus we added in the figure caption the sentence “The scale of time delay was calibrated with the oscillation frequency observed in the spectrum for the 6p Rydberg state”.

Comment: What limits the time resolution in this experiment? I suspect that the synchrotron pulses are long, and the wave packets can only be detected because the lifetimes of the fluorescent states are even longer than the pulse durations. Is yes, this severely limits the general applicability.

Answer: The concept of the coherent control achieved in this work relies on the use of longitudinal coherence within a light wave packet pair produced by individual relativistic electrons, rather than temporal coherence of a whole light pulse. Considering that every relativistic electron in an electron bunch travels in an identical magnetic field specifically shaped by a synchrotron device, all the light wave packets generated by the device has basically a common waveform (like the one illustrated in the right panel of Fig.1). We propose in this paper that coherent control by synchrotron radiation can be established by the use of the longitudinal coherence existing naturally in the waveform of the light wave packets. While temporal coherence of a whole light pulse is limited by the electron bunch length (300-ps duration at UVSOR), the common waveform that light wave packets have is essentially independent of the bunch duration. Thus, the temporal resolution achieved with the present coherent-control concept is completely unrelated to the bunch length and is determined by the accuracy and uniformity of the waveforms of light wave packet pairs. On the application in short-wavelength range, the light wave packet pairs become to have higher-frequency oscillations, but still the longitudinal coherence in the well-defined waveform is unaffected. Therefore, there is no essential restriction on application of the present coherent control concept to shorter wavelength. Here, the time range of target process is an interesting factor. In the present study, we observed nanosecond fluorescence in He, which is much longer than the time

scale of delay between the light wave packets. If we targeted a decay process much shorter than the minimum delay of 2.1 fs, the two wave packets launched inside the matter by the light wave packet pair could not interfere to each other and we would see no oscillation in the time spectrum. Most interesting case is that the decay lifetime is comparable to the time range we can scan (i.e. femtosecond range), where the envelope of the interference pattern will reflect the lifetime of the target state. Then one may consider that core-hole processes in x-ray range are too fast to be targeted. This may be true, but subsequent processes after the core-hole decays can be in the observation range. At present we cannot grasp how it appears on the time spectrum, and are interested in such study. We have already started the investigations to He doubly-excited states and Xe 4d resonances with a similar experimental scheme and a more advanced set-up (photoelectron spectroscopy), and publishable data will be gained soon.

To clarify these things, we inspected manuscript through and added/modified many sentences, particularly in pages 2&4.

Reviewers' comments:

Reviewer #1 (Remarks to the Author):

I thank the authors for their detailed response to each of the points raised during review. In the revised version the authors have explained in more detail the operating principle of this technique. The inclusion of additional measured spectra as supplementary material is very welcome.

The paper is suitable for publication.

Reviewer #2 (Remarks to the Author):

The authors have included most of my suggestions in a convincing way. Thus the manuscript should be published after a few improvements. In particular, it was not completely clear to me how the authors extract the relative populations of all states in Fig. 4. The authors could present a mathematical formalism in the supplementary material and give a more detailed description of their analysis.

Reply to the comments from Reviewer 2:

Thank you very much for the useful comments.

Comments: The authors have included most of my suggestions in a convincing way. Thus the manuscript should be published after a few improvements. In particular, it was not completely clear to me how the authors extract the relative populations of all states in Fig. 4. The authors could present a mathematical formalism in the supplementary material and give a more detailed description of their analysis.

Response: The time spectra in Fig. 3(b), showing the fluorescence intensities as a function of the time delay of the light wave packet pair, modulate with the individual frequencies of the Rydberg excitations. Thus the relative fluorescence intensities, which reflect the populations of the individual Rydberg states, vary according to the time delay we set. Fig. 4 plots simply the fluorescence intensities sampled at four particular time delays which are indicated in Fig. 3(b).

We realized from your comment that our explanation for this extraction of the Rydberg populations in Fig. 4 was unclear. We thus added some sentences in page 4 left and in the supplementary material.

In the supplementary material, to clarify the fitting function to the Ramsey fringe features in Fig. 3(b) and in Fig. S1, a description of the sinusoidal function and the corresponding explanation were added, as follows: “The time spectra in Fig. S1 (and in Fig. 3(b)), reflecting the populations of the corresponding Rydberg states, oscillate with individual frequencies. The population ρ of a Rydberg state at a time delay τ between the light wave packet pair is given by $\rho \propto (1 + \cos \omega\tau)$, where ω is the transition frequency from the ground state to the Rydberg state [1,16,17].”

In the text, the following sentences were added in page 4 left: “The time spectra in Fig. 3(b), modulating with the individual frequencies, manifest that the relative intensities of fluorescence from these Rydberg states are dependent on the time delay of the light wave packet pair. The relative intensities sampled at four particular time delays are represented in Fig. 4. It is clearly demonstrated here that the fluorescence intensities, reflecting the populations of these Rydberg states, strongly vary with the time delay.”

We hope that the extraction of the Rydberg populations shown in Fig. 4 is now sufficiently clear by these modifications in the text and the supplementary material.

REVIEWERS' COMMENTS:

Reviewer #2 (Remarks to the Author):

I thank the authors for their additional explanations. The manuscript should be published.